# Point Cloud Classification by Domain Adaptation Using Recycling Max Pooling and Cutting Plane Identification

**DOI:** 10.3390/s23031177

**Published:** 2023-01-19

**Authors:** Hojin Yoo, Kyungkoo Jun

**Affiliations:** 1Department of Embedded Systems Engineering, Incheon National University, Incheon 22012, Republic of Korea; 2Energy Excellence and Smart City Laboratory, Incheon National University, Incheon 22012, Republic of Korea

**Keywords:** point cloud classification, unsupervised domain adaptation, self-paced learning, self-supervised learning

## Abstract

Deep models have been studied in point cloud classification for the applications of autonomous driving and robotics. One challenging issue is that the point cloud of the same object could be discrepantly captured depending on sensors. Such a difference is the main cause of the domain gap. The deep models trained with one domain of point clouds may not work well with other domains because of such a domain gap. A technique to reduce domain inconsistency is domain adaptation. In this paper, we propose an unsupervised domain adaptation with two novel schemes. First, to improve unreliable pseudo-label assignment, we introduce a voting-based procedure based on the recycling max pooling module, which involves self-paced learning. It helps to increase the training stability of the models. Second, to learn the geometrical characteristics of point clouds in unfamiliar settings, we propose a training method of cutting plane identification, which works in an unsupervised way. Testing with the popular point cloud dataset of PointDA-10 and Sim-to-Real, experiments show that our method increase classification accuracy by 6.5%-points on average, ModelNet and ShapeNet as the source domain and ScanNet, and ScanObjectNN as the target domain. From an ablation study, it was observed that each method contributes to improving the robustness of domain adaptation.

## 1. Introduction

The most straightforward way to depict a 3D object is as a point cloud. Deep learning techniques that use the point cloud to learn classification and segmentation [1,2,3,4,5,6,7,8], detection [9,10], or completion [11,12] are currently being researched. The ability of the deep model to learn these tasks is crucial in the fields of autonomous driving and robotics.

The point cloud can be constructed by gathering sensor data from real-world objects using a depth camera or Light Detection and Ranging (LiDAR). The point cloud can also be created by sampling a 3D shape. The points in the synthetic point cloud, which is produced in virtual space, are dispersed uniformly, and the data may be easily updated or modified. On the other hand, a real-world point cloud’s distribution might not be regular due to equipment noise, and an object might not be entirely made up of obstacles. Furthermore, a single object’s point cloud may be captured differently depending on the sensors used. From this perspective, it is considered that a domain shift takes place when attempting to develop the deep model using these point clouds.

The deep model must be adjusted to the environment in which it is intended to work in order to overcome domain shift. However, gathering data is a labor-intensive operation that takes a lot of time. Alternatively, synthetic point clouds could be produced with simplicity utilizing a Computer-Aided Design (CAD). A source domain is an environment rich in labeled data. A target domain is an environment where the deep model functions but where labeled data are scarce. Unsupervised domain adaptation (UDA) should be employed in training to enhance the deep model’s performance in the target domain.

There are two basic ways to apply UDA to point clouds, according to recent studies. The first technology utilizes adversarial learning, which makes it harder to distinguish apart different domains [13]. Tasks can be completed regardless of the domain because this technique decreases the deep model’s ability to discriminate input domains. The second method makes use of self-supervised learning, which teaches the geometrical properties of the domain on its own [14,15,16,17,18]. Self-supervised learning is an approach for establishing deep models about the properties of the domain so that tasks can be carried out in the domain. The strategy using self-supervised learning is currently the one with the best performance. Geometry-Aware Self-Training (GAST) [14], which encodes features by identifying angle and distorted parts of point clouds, was proposed.

However, the GAST’s pseudo-label generation mechanism is unreliable because the pseudo-label is determined by a single prediction result, which has the highest confidence score. It ignores other prediction results even though their scores are high, resulting in wrong label assignments. Research on UDA of the point cloud is also lacking as compared to 2D image fields, where several self-supervised learning methods, such as predicting positional relationships of partial images and predicting the order of partial ones, are examined. Therefore, we suggest voting-based self-paced learning with a recycling max pooling module and unprecedented self-supervised learning method illustrated in Figure 1.

Therefore, this paper introduces a novel UDA method that could be applied to point clouds. Our contributions are summarized as follows.

We propose a novel voting-based procedure based on recycling the max pooling module and using improved cross-entropy loss function.We suggest a training method of cutting plane identification, which is a novel self-supervised learning method for domain adaptation. A corresponding cross-entropy loss function was also presented for learning.We showed that the proposed methods improve point cloud classification accuracy through ablation studies, comparison studies with existing methods, and additional studies.

## 2. Related Work

**Point Cloud Classification–**There are two sorts of recent point cloud classification networks. The first one is a method for understanding the point cloud’s characteristic features using multi-layer perception (MLP), which can represent the association between all points [1,2,5,7]. This tactic includes a methodology for encoding all information between points into the MLP. The method has a weakness in that the deep model could not recognize local features due to a lack of regional information. An approach using local features with surrounding points [19] and was using a graph with those local features [3,4,6,8] were investigated to address the drawbacks. These networks tried to learn global or regional geometric features through point cloud classification. The domain adaptation on the point cloud has not been considered.

**Unsupervised Domain Adaptation for Point Cloud–**An UDA method used in a 2D image field could be classified into two types. In the first scheme, domain adaptation is attempted by reducing domain disparities. This approach is broken down into two parts: a way to utilize a statistical distribution and a technique to leverage domain distance. Minimizing the maximum mean discrepancy (MMD) between the source domain and the target domain is a representative way to increase classification accuracy [20,21]. The second way aligns features across domains is by employing adversarial learning. The method for aligning domains with a domain classifier and an inverse gradient layer has been proposed [22,23].

In a recent point cloud study, domain adaptation was attempted using both a method of minimizing domain difference and adversarial learning. Qin et al. [13] proposed a node module that could learn partial features of the point cloud and a way to minimize MMD between nodes. At the same time, adversarial learning was also applied to take in overall characteristics between domains. The amount of information the point cloud contains when learned partially is limited. Therefore an optimal local issue may arise.

**Self-supervised Domain Adaptation for Point Cloud–**Self-supervised learning is the process of exploiting input data that has been transformed into a format suitable for supervised learning. Through self-supervised learning, deep learning models could explore what the dataset represents. Recent 2D image research has suggested learning representativeness through tasks including predicting the positional relationship of two parts in an image [24], predicting the rotation angle of an image [25], and predicting order using a jigsaw puzzle [26]. These works have demonstrated that when supervised learning is implemented after comprehending the representation of images via self-supervised learning, the performance of deep models may be enhanced.

Studies have sought to increase classification accuracy in point clouds by adopting self-supervised learning. Sauder [15] used a method to predict position after dividing a point cloud into voxel units, and Poursaeed et al. [16] tried to improve classification accuracy by rotating the entire point cloud and then predicting the rotated angle. Recently, researchers have tried to acquire the geometrical properties of point clouds that use self-supervised learning in an effort to adapt the domain. In Achituve et al. [17], a point cloud is partially distorted and then reconstructed. Zou et al. [14] suggested a methodology for self-supervised learning that predicts the rotation angles of two-point clouds and distorted positions. In addition, in Shen et al. [18], the reconstruction of the point cloud is carried out with a feature map of the point cloud and sampled points.

These strive at domain adaption employing self-supervised learning on point clouds. There have been relatively few studies on self-supervised learning algorithms for point clouds since, unlike 2D images, they cannot be described simply using coordinates. For example, the self-supervised learning method using a jigsaw puzzle could not be easily implemented since the three-dimensional coordinate is more intricate than the two-dimensional coordinate. Therefore, research is required into how to effectively employ point clouds as self-supervised learning data.

## 3. Unsupervised Domain Adaptation for Point Cloud Classification

Source domain labeled point clouds S=P𝒾s,𝓎𝒾s𝒾=1ns and target domain unlabeled point clouds T=P𝒾t𝒾=1nt are utilized in UDA on point cloud classification. The point cloud P∈X⊂ℝN×3 consists of N three-dimensional coordinated x,y,z. A classification result is obtained by Equation (1).
(1)ΦP=ΦclsΦfeaP
where Φfea creates features of the input point cloud, and Φcls which is composed of fully connected (FC) layers and classifies a category of the point cloud. The equation of the classification deep model Φ in GAST [14] has been replaced as follows in Equation (1) because. The reason is that, in our opinion, the output of Φfea is used as the input of Φcls.

Figure 2 depicts the suggested UDA procedure. It consists of three main components. The multiple one-dimensional feature vectors from the point cloud are produced by the feature extractor $\Phi_{fea}$ and recycling max pooling module. The category classifier $\Phi_{cls}$ identifies the class of the point cloud, and the cut plane classifier $\Phi_{cut}$ distinguishes the direction of cutting planes. We propose a novel UDA that makes use of self-paced learning and self-supervised learning. Before providing the proposed method, Section 3.1 explains the GAST learning category classification and domain adaptation method. Section 3.2 then introduces the proposed self-paced learning method. Finally, Section 3.3 illustrates the suggested self-supervised learning strategy.

### 3.1. Domain Adaptation on Geometry-Aware Self-Training

#### 3.1.1. Learning Category Classification

For the point cloud ${\left\{P_{i}^{s},{𝓎}_{i}^{s}\right\}}_{i=1}^{n_{s}}$ in the source domain, the model Φ predicts a category with ${\left\{p_{i}^{s}\right\}}_{i=1}^{n_{s}}$, category classification learning in the source domain is conducted using cross-entropy loss, as shown in Equation (2).
(2)$$L_{cls}=-\frac{1}{n_{s}}\overset{n_{s}}{\underset{i=1}{\sum }}\overset{C}{\underset{c=1}{\sum }}I\left[c={𝓎}_{i}^{s}\right]\log p_{i,c}^{s}$$
where $p_{i,c}^{s}$ means c-th predicted value of ${\left\{p_{i}^{s}\right\}}_{i=1}^{n_{s}}$, and $I\left[\cdot \right]$ means indicator function. $I\left[c={𝓎}_{i}^{s}\right]$ means a value in that class. It has a value of 0 or 1, and one value of 1 per object $n_{s}$.

Since there are no labeled data, category classification learning in the target domain uses self-paced learning. Creating pseudo labels based on the model’s computed prediction results and applying them for training is known as self-paced learning. For the point cloud ${\left\{P_{i}^{t}\right\}}_{i=1}^{n_{t}}$ in the target domain, the model Φ predicts a category ${\left\{p_{i}^{t}\right\}}_{i=1}^{n_{t}}$. The category with the highest predicted value is taken into consideration as the label, and a pseudo-label is formed. When the label of the corresponding category is ${\hat{𝓎}}_{𝑖,c}^{t}$, a pseudo label is brought about in Equation (3).
(3)$${\hat{𝓎}}_{𝑖,c}^{t}=\left\{\begin{array}{cc} 1, & \;\;if\;c=\arg\underset{c^{\prime }}{\max}p_{i,c^{\prime }}^{t},p_{i,c}^{t}>\exp\left(-\gamma \right)\\ 0, & \;\;otherwise. \end{array}\right.$$
where $p_{i,c}^{t}$ means c-th predicted value of ${\left\{p_{i}^{t}\right\}}_{i=1}^{n_{t}}$.

It is considered that producing pseudo labels with only the highest value is an unreliable method, so the methods that could generate reliable pseudo labels have been studied. Fan et al. [27] proposed a voting-based pseudo-label generation method using the k-nearest neighbors (k-NN) algorithm. However, it has the shortcoming that pseudo labels can be determined only after all the feature vectors of input data are fed into the k-NN algorithm, unable to be used during the deep model training in real time. Our proposed method addresses this problem by having the pseudo-label determination process run during the model training at the same time.

#### 3.1.2. Self-Supervised Learning on Geometry-Aware Self-Training

Through encoding geometrical features of point clouds to deep models using two approaches, domain adaptation by self-supervised learning was attempted in GAST. The first method is rotation angle identification (RotCls), and the other method is distortion area identification (LocCls). The identification results are trained using cross-entropy loss in both methodologies.

The angle is the rotated point cloud is anticipated throughout the RotCls phase. In the beginning, the point cloud P is divided into two-point clouds $P_{a}$ and $P_{b}$. Then, $P_{a}$ is rotated along x-axis clockwise and $P_{b}$ is rotated along y-axis in clockwise angle within {0°, 90°, 180°, 270°}. A new point cloud $P_{rot}$ which would be used as input for a deep model consisting of $P_{a}$ and $P_{b}$ rotated in x-axis and y-axis. Using $P_{rot}$ as input, $P_{a}$ and $P_{b}$ are each identified in rotation angle.

LocCls refers to the process of distorting a portion of the point cloud and predicting a number of distorted regions. First, P is separated into $k^{3}$ regions in units of voxels, and then one section is selected for distortion. A new point cloud $P_{loc}$ is created by applying distortion by sampling with an isotropic Gaussian distribution around the point is selected one point in the chosen area. Using $P_{loc}$ as input, which of the $k^{3}$ region is distorted is classified.

### 3.2. Preliminary: Recycling Max Pooling

The backbone network of the point cloud deep model creates a feature matrix expressing correlation for each point or graph. By utilizing max pooling on the matrix, a feature vector is produced, after which classification or segmentation is carried out using the feature vector’s information. In order to improve the performance of the point cloud deep model, a method of utilizing the feature matrix has been studied.

Chen et al. [28] argued that a lot of information is discarded from the feature matrix in the max pooling. Chen et al. suggested a recycling max pooling module that extracts the second and third max pooling vectors in addition to the first max pooling vector. The experiment was presented, showing similar performance even when the classification was performed using the second or third feature vector. This showed that not only the first max pooling vector but also the subsequent max pooling vectors contributed, have significant values. By using these sequential max pool processes, we can extract multiple label candidates from a single input, contributing to the improved label determination scheme. All feature vectors generated by the recycling max pooling module had their loss values calculated and were reflected in the deep model. Additionally, it reveals that the accuracy increased when the scheme was used with various deep-point cloud models.

### 3.3. Proposed Self-Paced Learning: Voting Pseudo Label with Recycling Max Pooling

The recycling max pooling is applied to the following procedure to produce multiple feature vectors. When point cloud P passes through the feature extractor Φ, the feature matrix P1f∈ℝN1×M can be obtained. N1 denotes the number of points before the first max pooling, and M means the dimension of the feature matrix. After the first max pooling is performed, we will gain a feature vector F1∈ℝM and a feature matrix P2f∈ℝN2×M which excludes the feature vector F1. N2 implies the number of points after the first max pooling vector is excepted. The process is repeated r times until the rth feature vector Fr∈ℝM is earned.

r feature vectors will be utilized for supervised learning of the source domain point cloud, self-paced learning of the target domain point cloud, and self-supervised learning. Section 3.3.1 introduces the procedure of learning category classification using the feature vectors of the source domain point cloud. The issue with the existing pseudo-generating method is brought up in Section 3.3.2, and a revised approach that makes up for the disadvantage while presenting the voting procedure is suggested in Section 3.3.3.

#### 3.3.1. Supervised Learning in Source Domain with Recycling Max Pooling

When the point clouds in the source domain are given such as ${\left\{P_{i}^{s},{𝓎}_{i}^{s}\right\}}_{i=1}^{n_{s}}$, r feature vectors can be obtained for one point cloud. Each feature vector will pass through the category classifier $\Phi_{cls}$ to obtain a category prediction result $p_{i}^{s}$. The loss is calculated with a cross-entropy function using the prediction results as shown in Equation (4) by obtaining the average of the r feature vectors.
(4)$$L_{cls}^{s}=-\frac{1}{n_{s}}\overset{n_{s}}{\underset{i=1}{\sum }}\frac{1}{r}\overset{r}{\underset{j=1}{\sum }}\overset{C}{\underset{c=1}{\sum }}I\left[c={𝓎}_{i}^{s}\right]\log p_{i,j,c}^{s}$$

$p_{i,j,c}^{s}$ means c-th predicted value. The deep model may learn to be more generalized than when employing a single feature vector because learning happens with the outcomes from several feature vectors.

#### 3.3.2. Problem of Existing Self-Paced Learning

The pseudo-label generation method of GAST has a disadvantage that could happen while determining the learning direction. The reason is that the deep model recognizes the category with the highest score as the answer. If the deep model sees one object as another category at the beginning of learning and generates an incorrect answer, the issue that the deep model keeps learning with the incorrect label emerges. As a result, classification performance for the target domain point cloud suffers.

In order to reimburse the drawback, we propose the pseudo-label generation method using voting with the recycling max pooling module. In the previous study, the pseudo label was determined by examining just one classification result because the deep model of GAST makes one-dimensional output. Ours uses votes from a variety of results to choose the label. The dependability of the pseudo label is projected to be increased by this method, which is anticipated to be crucial in establishing the learning direction.

#### 3.3.3. Pseudo Label Generation with Voting

Figure 3 illustrates the voting-based pseudo-label generation mechanism leveraging the recycling max pooling module. When the target domain point cloud $P_{i}^{t}$ passes through the feature extractor $\Phi_{fea}$, recycling max pooling module, and the category classifier $\Phi_{cls}$, r category prediction results can be obtained. As shown in Equation (5), the label is determined through voting.
(5)$${\tilde{𝓎}}_{i}^{t}=vote\left(\left\{{𝓎}_{j}^{t}|j\in {\left\{p_{i}^{t}\right\}}_{i=1}^{r}\right\}\right)$$

${\tilde{𝓎}}_{i}^{t}$ means a pseudo-label vector generated through voting and is in the form of a one-hot vector. The label is created only when the proportion of the results of a specific category is greater than a threshold $\lambda_{vote}\in \left[0,1\right]$ and learning is performed as shown in Equation (6) through the cross-entropy loss. Self-paced learning will not be used for the remaining point clouds.
(6)$$L_{cls}^{t}=-\frac{1}{n_{t}}\overset{n_{t}}{\underset{i=1}{\sum }}\frac{1}{r}\overset{r}{\underset{j=1}{\sum }}\overset{C}{\underset{c=1}{\sum }}I\left[c={\tilde{𝓎}}_{i}^{t}\right]\log p_{i,j,c}^{t}$$

The category classification loss value of the source domain $L_{cls}^{s}$ introduced in Section 3.3.1 is combined with $L_{cls}^{t}$ as in Equation (7) to proceed with the point cloud category classification learning.
(7)$$L_{sem}=L_{cls}^{s}+\lambda_{t}L_{cls}^{t}$$

$\lambda_{t}\in \left[0,1\right]$ is a penalty parameter for reducing noise at the beginning of the learning.

### 3.4. Proposed Self-Supervised Learning: Cutting Plane Identification

We present a novel method for the model to learn geometric information and cutting plane identification (CutCls). As shown in Figure 4, CutCls entails dividing the point cloud in half on a plane in either the x-z or y-z direction and then forecasting the cut direction between the two planes. This approach was motivated by the idea that geometric features could be learned by predicting the cut plane axis of the point cloud. We believe that this technique could extract and encode overall geometrical characteristics into feature vectors. The training process involving the classification of the cut direction x-z or y-z forces the model to learn geometric features from an unfamiliar environment. As a result, the backbone of the model can extract point cloud features even from samples of different domains.

Moreover, the CutCls are expected to be more effective in using a cut point cloud, unlike traditional methods. For instance, it is challenging to find an optimal value in the case of the LocCls of GAST since it is a problem to predict the answer among $k^{3}$ classification. On the other hand, CutCls has the advantage of being able to use self-supervised learning that performs binary classification to quickly converge desired geometric feature learning. The binary classification means that the statement is determined between two values. It is possible to rapidly reach the optimal value because there is less information to learn.

When a set of x-axis or y-axis coordinate values of the plane to be cut in the point cloud, P is ${\left\{o_{i}^{m}\right\}}_{i=1}^{m}$, cutting coordinate is decided as in Equation (8).
(8)$$CutCoord=\frac{\underset{o}{\max}{\left\{o_{i}^{m}\right\}}_{i=1}^{m}-\underset{o}{\min}{\left\{o_{i}^{m}\right\}}_{i=1}^{m}}{2}$$

If the point cloud P is cut along the selected plane from the obtained coordinates, two-point clouds $P_{l}$ and $P_{r}$ could be gained. Which of the point cloud should be employed is decided randomly because if only one of the two point clouds is used as input, self-supervised learning may not be successfully carried out. The deep model will have learning biases toward one class, and it will cause class imbalance.

The selected point cloud cut in half needs to be adjusted the number of points. A point cloud has a variable distribution of points for each point cloud, so cutting in the following method will produce a different number of points. On the other hand, if a 2D image were chopped in half, the quantity of information would be exactly halved. Therefore, the number of points should be sampled as Pcut∈ℝm2×3 through farthest distance sampling used in PointNet [1].

The dataset created for domain adaptation could be indicated as ${\left\{P_{cut,i},h_{i}\right\}}_{i=1}^{n_{s}+n_{t}}$. $h_{i}$ means the direction of the cut plane and has the form of a one-hot vector. Using the point cloud $P_{cut}$ as input, the direction of the cut plane is predicted among x-z plane or y-z plane through $\Phi_{cut}$. When the set of predicted results is ${\left\{p_{i,j,g}\right\}}_{i=1}^{n_{s}+n_{t}}$, the model uses cross-entropy loss as in Equation (9) to encode geometric features.
(9)$$L_{cut}=-\frac{1}{n_{s}+n_{t}}\overset{n_{s}+n_{t}}{\underset{i=1}{\sum }}\frac{1}{r}\overset{r}{\underset{j=1}{\sum }}\overset{G}{\underset{g=1}{\sum }}I\left[g=h_{i}\right]\log p_{i,j,g}$$

Domain adaptation is conducted by summing $L_{cut}$ and $L_{sem}$ which was gained from Section 3.3.3 as in Equation (10).
(10)$$L_{total}=L_{sem}+\lambda_{cut}L_{cut}$$
where $\lambda_{cut}$ is trade-off hyper-parameter.

## 4. Experiments

### 4.1. Datasets

**PointDA–**The PointDA [13] is used to check whether domain adaptation is performed. PointDA is a dataset that collects point cloud data corresponding to 10 categories common to ModelNet40 [29], ShapeNet [30], and ScanNet [31]. Only objects in the 10 categories are collected in ModelNet40 and are known as ModelNet-10 (M10). ShapeNet-10 (S10) is from ShapeNet, and ScanNet-10 (S*10) is from ScanNet. Table 1 shows the number of training and tests each dataset has and the property of datasets.

M10 and S10 are data obtained by uniformly sampling points from a 3D model artificially created through 3D CAD. However, S*10 has data acquired by separating corresponding to specific objects from the point cloud collected by a depth camera in real space. For this reason, ScanNet’s point distribution is not uniform, in contrast to ModelNet40 and ShapeNet. Additionally, since depth information is not partially gathered, some point clouds may not accurately depict the object.

**Sim-to-Real–**The Sim-to-Real [32] is a new benchmark, which shares 11 categories with ModelNet40 and shares nine categories with ShapeNet from ScanObjectNN [33]. The dataset is gathered to estimate the effectiveness of meta-learning on point clouds. Our approach, which applies point cloud domain adaptation, is evaluated with the dataset. The dataset has four subsets: ModelNet-11 (M11), ScanObjectNN-11(S*O11), ShapeNet-9 (S9), ScanObjectNN-9 (S*O9).

### 4.2. Experiment Setup

**Comparative Methods–**The dataset was tested only when the target domain were real-world datasets, such as ScanNet-10, ScanObjectNN-11, and ScanObjectNN-9. The reason is that in order to check if this technology could be used in the real-world, data of the target domain should be from the real-world. We conduct experiments with four types of scenarios. PointDA: M10 → S*10, S10 → S*10, Sim-to-Real: M11 → S*O11, S9 → S*O9.

For evaluation, it was examined whether the proposed method improves the accuracy compared to the case without domain adaptation. In addition, it was also investigated how the accuracy was improved when combined with the two ways (LocCls and RotCls) proposed by GAST [14]. The comparison with the GAST method was inspected on our platform, and the accuracy of the GAST was referred to for supervised learning and remaining domain adaptation methods.

**Implementation Details–**Backbone network $\Phi_{fea}$ for extracting the point, cloud features are based on DGCNN [8] proposed in 2019. The DGCNN is an improved deep model for point cloud than PointNet [1] or PointNet++ [2], and its benchmark is higher. In addition, The DGCNN is most recently used for comparison in the field of point cloud domain adaptation research. The category classifier $\Phi_{cls}$ has three FC layers of MLP, each having 512, 256, and 10 nodes, respectively. The model predicts 10 categories of point clouds. A classifier for cutting plane prediction self-supervised learning uses one FC layer. The dimension of our point cloud dataset is all set to 1024.

As in GAST, z-axis direction random rotation data augmentation was applied, the Adam [34] optimizer was utilized, the initial learning rate was 0.001, the weight decay was 0.00005, and a cosine annealing learning rate scheduler was employed. All experiments were performed for 200 epochs with a batch size of 16. The hyperparameter of $\lambda_{cut}$ is set to 1. The r is set to 5. The experiment was carried out after the most credible label had been produced with $\lambda_{vote}$ being set to 1. The test was performed by measuring the classification accuracy without data augmentation on the test set of each data set. In total, we train all the methods with the same random seeds on an NVIDIA Geforce RTX 3070 GPU.

### 4.3. Results

The experiments were conducted by classifying them into three purposes. First, the ablation study signifies that each of the proposed methods is effective. Second, the comparison studies show how much performance improvement was achieved when compared to the domain adaptation methods that were previously applied. Finally, in additional studies, we analyze how changing different hyperparameters affects accuracy.

#### 4.3.1. Ablation Studies

The ablation studies were performed to see if the proposed self-paced learning technique and self-supervised learning method are effective for domain adaptation. For the two scenarios M10 → S*10 and S10 → S*10, the accuracy without applying the domain adaptation methods was calculated, and the effectiveness of each method was proved. The accuracies of the experiments are shown in Table 2.

In the M10→S*10 scenario, both self-paced learning and cutting plane identification show improved accuracy than when there is no domain adaptation. In addition, when learning by combining the two methods, better accuracy was recorded than when each was applied, with a 9.5%-point increase. When self-paced learning was used in the S10→S*10 scenario, the accuracy was improved. However, the accuracy decreased when the CutCls were used alone.

We had held the opinion that the ShapeNet-10′s data imbalance was to blame for the result because it has a disproportionate amount of data in some categories. For example, chair and table point clouds accounted for 60.4% of the dataset. Therefore, we evenly adjusted the number of samples in each category and executed the experiment in same scenario. The results are different from our expectations; the accuracy does not improve at all. It even drops. It means the class imbalance was not the cause of the dropped accuracy.

To investigate the reason from another perspective other than the class imbalance, we analyzed the classification accuracy for each category as shown in Table 3. As a result, we find that 3 out of 10 categories are responsible for lowered accuracy (18% points drop in average); they are bathtub, sofa, and table, while the accuracy of other four categories improves, and the remaining categories do not change. The common characteristic of the three categories is that they are all relatively simple and big objects, while the improved four categories are bookshelf, lamp, monitor, and plant, which are small and have a complex shapes.

Furthermore, when the two methods were put on learning at the same time, the accuracy increased by 8.9%-points. This finding demonstrates that each suggested strategy for domain adaptation affects classification in the target domain.

#### 4.3.2. Comparison Studies

The accuracy of the four scenarios introduced in Section 4.2 is assessed to see if the proposed self-paced learning and cutting-plane self-supervised learning approaches are successful. The accuracy results are compared with the accuracy of the existing method. Furthermore, we observed the outcomes of integrating our method with the existing self-supervised learning approach. LocCls and RotCls of GAST were merged with the CutCls, respectively, and an experiment combining all three was also conducted. Table 4 shows the results of the comparison studies.

In the M10 → S*10 and S10 → S*10 scenarios, using the existing GAST method increased by 8.5%-points and 6.6%-points, respectively. When the proposed method was exploited, the accuracy separately rose by 9.5%-points and 8.9%-points. Even in the M11 → S*O11 and S9 → S*O9 scenarios, the accuracy rather decreased when learning with existing domain adaptation. Conversely, when we employed our strategy, the accuracy was enhanced by 5.0%-points and 2.4%-points, respectively.

It is presumed that the cause of the diminished accuracy with the existing method in the M11 → S*O11 and S9 → S*O9 scenarios is the problem with the self-paced learning method. When generating a pseudo label in the self-paced learning of the GAST, the category with the highest score in the predicted value is created as a label to learn. If an object’s label is produced wrongly at the start of learning, there’s a chance that the inaccurate label will continue to be learned later.

On the one hand, the characteristics of ScanNet-10 are markedly different for each category. On the other hand, in ScanObjectNN-11, there are categories with overlapping features. For instance, in bed and desk or cabinet and door, the shape of these objects looks similar when visualized in 3D. Studies would suffer in the long term if a model confused these objects to produce pseudo labels. An example of the objects, each having similarities, is shown in Figure 5.

The proposed voting-based pseudo-label generation method could reduce confusion among similar objects. This is because the pseudo label is not used for learning if the voting rate falls short of the threshold. Since learning proceeds only when the deep model is confident in the category, it can be expected that the accuracy is improved by generating steady pseudo labels. Table 5 shows that the accuracies of some categories increased when our proposed method was utilized. For the two categories of desk and bed, the accuracy improved by 20% points on average, proving that our voting scheme is effective, while the results of the door were similar. However, in the cabinet category, the accuracy dropped from 8.5% to 0%. We carefully investigated the cabinet samples from two datasets, ModelNet and ScanObjectNN. Surprisingly, we found that the cabinets are quite different in the two datasets, as shown in Figure 6. For example, the cabinets of ModelNet look similar to desks, while those of ScanObjeectNN look similar to wardrobes. Such a difference is likely to be the main cause of the low accuracy.

When learning in blend with GAST’s self-supervised learning method, the highest accuracy was recorded if RotCls was merged in the M10 → S*10 scenario. In addition, in the S9 → S*O9 scenario, the highest accuracy is accounted for when all methods are combined. On the other hand, when LocCls was mingled, the accuracy was higher than the state without adaptation, but it did not record the highest accuracy. This shows that the performance improvement of the LocCls is insignificant compared to the other methods. The reason for this phenomenon is that the learning process of LocCls is more complicated than others.

#### 4.3.3. Additional Studies

In order to examine the impact of the pseudo label generation threshold value on learning, an experiment was run while varying the value of $\lambda_{vote}$. In the M10 → S*10 scenario, when the $\lambda_{vote}$ value was 1.0, 0.8, and 0.6, respectively. The proposed method was used, and the accuracies were compared. As shown in Table 6, the learning results show that the accuracy is highest when the $\lambda_{vote}$ value is 1.0. In addition to that, the lower $\lambda_{vote}$ value, the greater drop in accuracy. It is speculated that this is because the false label occurs more frequently as the threshold value decreases.

Moreover, an experiment was conducted to investigate how the accuracy change with the r value or recycling max pooling number. In the M10 → S*10 scenario, when the value was sequentially increased to 3, 5, 15, and 20, the change in accuracy was checked. The learning results are shown in Table 7. The findings indicate that when the r value rises, accuracy may also rise. Since there are fewer recycling max pooling and the voting dependability is worst, the accuracy is lowest when r=3. The accuracy is maximum when the r value is 15, but it is lower than 15 when the value is 20. This suggests that overfitting occurred because there were too many recycled vectors employed for learning. Although the accuracy is the highest when the value is 15, the inference time of the deep model will increase because of the rise of r value. Therefore, it is important to choose a value that takes operational time into account.

For the purpose of researching the effect of the angle of the cutting plane, an experiment was handled by adjusting the cutting plane. The cutting plane was rotated by {10°, 30°, 45°} in y-axis direction, respectively, and then cut to generate $P_{cut}$. The result of the experiment is shown in Table 8. It can be seen that the highest accuracy is achieved when the cutting plane was without rotation in the M10 → S*10 scenario. Since the input point’s angle is not rotated in the y-axis when we test it, we anticipate that this is the reason. This implies that the optimal domain adaptation strategy is cutting the plane vertically.

## 5. Conclusions

In this paper, we proposed a novel unsupervised domain adaptation method applicable to point cloud classification. The novel voting-based procedure based on the recycling max pooling module was presented. In addition, we introduced the self-supervised learning method and raised the possibility of expanding the self-supervised learning in point cloud classification.

According to the experiments, the proposed methods were helpful for the domain adaptation of point clouds. Each strategy improved accuracy, it was found through an ablation study. Furthermore, the most appropriate mechanism for domain adaptation was discovered by investigations combined with the existing approaches. Finally, the optimal conditions were known via additional studies adjusting the hyperparameters of the proposed method.

The suggested point cloud domain adaptation method does, however, have certain limitations. When generating pseudo labels with voting on multiple vectors, the problem will occur when the voting rate does not exceed the threshold. Our self-paced learning process was abandoned if the rate failed to surpass the threshold. In other words, the dataset’s information was not being utilized to its full potential. If adversarial learning or other pseudo-label generation methods can be used for the data in this situation, higher accuracy can be expected. We will investigate this approach in future work.

## Figures and Tables

**Figure 1 sensors-23-01177-f001:**
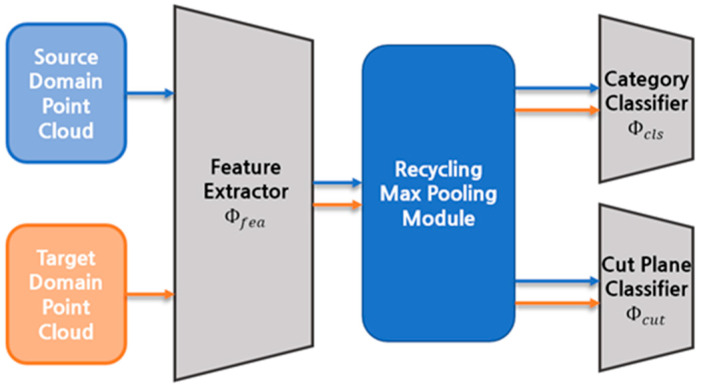
Concept of our domain adaptation method on point cloud classification.

**Figure 2 sensors-23-01177-f002:**
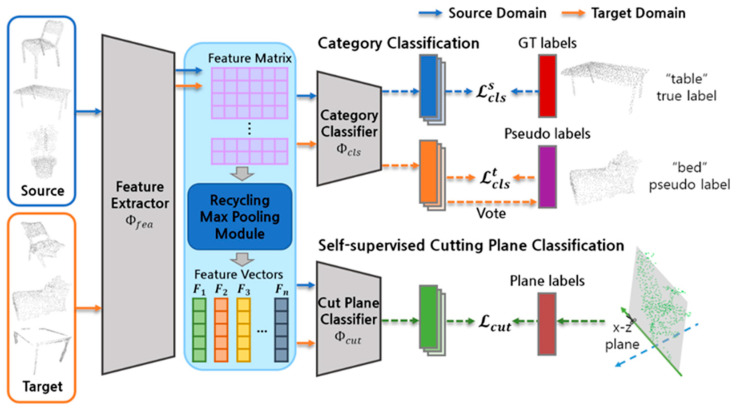
Overview of the proposed domain adaptive point cloud classification model.

**Figure 3 sensors-23-01177-f003:**
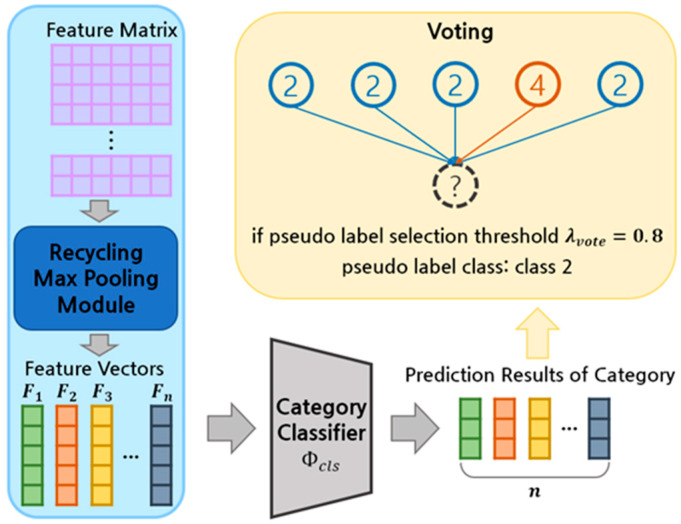
Pseudo-label voting process utilizing recycling max pooling module.

**Figure 4 sensors-23-01177-f004:**
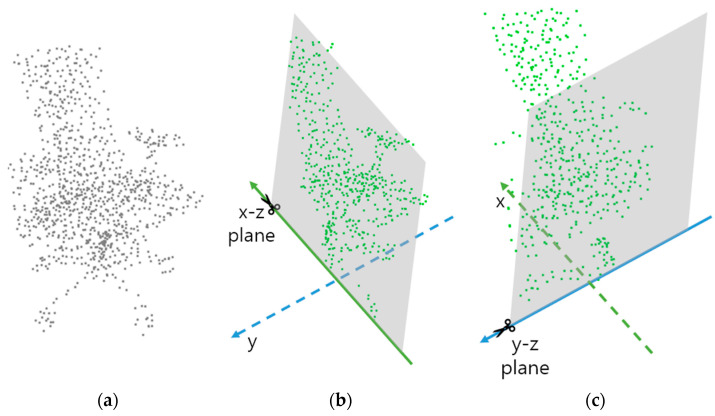
Illustrative examples of generating a cutting plane identification point cloud. (**a**) Original point cloud. (**b**) Point cloud cut on x-z plane. (**c**) Point cloud cut on y-z plane.

**Figure 5 sensors-23-01177-f005:**
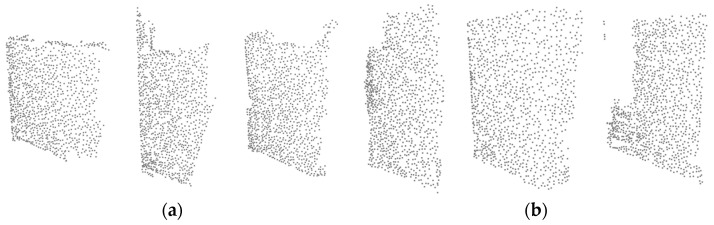
Examples of object categories with similar characteristics in ScanObjectNN-11. (**a**) Cabinet. (**b**) Door. (**c**) Bed. (**d**) Desk.

**Figure 6 sensors-23-01177-f006:**
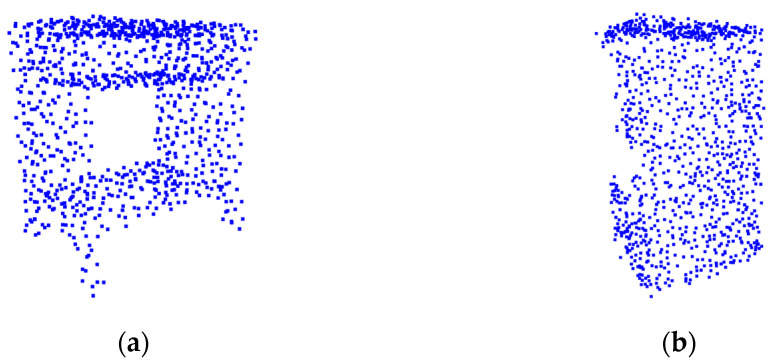
The example of a cabinet pointcloud of ModelNet and ScanObjectNN. (**a**) ModelNet. (**b**)ScanObjectNN.

**Table 1 sensors-23-01177-t001:** Characteristics and compositions of the dataset.

Dataset Name	Characteristic	Num. Train	Num. Test
ModelNet-10	Synthetic	4183	856
ShapeNet-10	Synthetic	17,378	2492
ScanNet-10	Real-world	6110	2048

**Table 2 sensors-23-01177-t002:** Accuracy (%) of ablation study on M10→S*10 and S10→S*10 scenario.

Adaptation Method		M10→S*10	S10→S*10
Voting SPL	CutCls		
		46.8	47.5
✓		53.9	55.1
	✓	49.6	46.4
✓	✓	56.3	56.4

**Table 3 sensors-23-01177-t003:** Category-wise classification accuracy (%) on the S10 → S*10 scenario.

	Bathtub	Bed	Bookshelf	Cabinet	Chair	Lamp	Monitor	Plant	Sofa	Table
w/o Adapt	61.5	0	0.1	0.1	63.9	41.5	44.3	0	38.1	65.8
Only CutCls	42.3 (−19.2)	0 (0)	23.3 (+23.2)	0 (−0.1)	65.8 (+1.9)	58.5 (+17.0)	60.7 (+16.4)	64.0 (+64.0)	26.1 (−12.0)	45.5 (−20.3)

**Table 4 sensors-23-01177-t004:** Accuracy (%) on the PointDA and Sim-to-Real datasets. The accuracy for models marked with † is tested on our platform.

	M10→S*10	S10→S*10	M11→S*O11	S9→S*O9
Supervised	78.4	78.4	-	-
w/o Adaptation†	46.8	47.5	58.8	46.1
DANN [35]	42.1	50.9	-	-
PointDAN [13]	44.8	45.7	-	-
RS [36]	46.7	51.4	-	-
DefRec+PCM [17]	51.8	54.5	-	-
GAST† [14]	55.3	54.1	57.1	41.9
Ours†	56.3	**56.4**	**63.8**	48.5
Ours + LocCls†	55.2	54.7	61.2	48.1
Ours + RotCls†	**57.3**	55.8	61.3	47.9
Ours + LocCls + RotCls†	56.5	55.2	58.9	**48.6**

**Table 5 sensors-23-01177-t005:** Part of category-wise classification accuracy (%) on the M11→S*O11 scenario.

	**Desk**	**Bed**	**Door**	**Cabinet**
GAST [14]	39.0	33.3	99.5	8.3
Ours	76.6	45.2	97.6	0

**Table 6 sensors-23-01177-t006:** Accuracy (%) of pseudo label generation threshold change experiment on M10 → S*10.

$\lambda_{vote}=1.0$	$\lambda_{vote}=0.8$	$\lambda_{vote}=0.6$
**56.3**	50.7	50.2

**Table 7 sensors-23-01177-t007:** Accuracy (%) of the number of recycling max pooling change experiments on M10 → S*10.

r=3	r=5	r=15	r=20
49.8	56.3	**56.4**	53.4

**Table 8 sensors-23-01177-t008:** Accuracy (%) of cutting plane angle rotation experiment on M10 → S*10.

0°	10°	30°	45°
**56.3**	52.2	52.5	52.8

## Data Availability

Not acceptable.

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
