# Peer review of "Point Cloud Classification by Domain Adaptation Using Recycling Max Pooling and Cutting Plane Identification"

_sensors, 2023, doi:10.3390/s23031177_

Round 1

Reviewer 1 Report

This paper proposed two schemes of unsupervised domain adaptive point cloud training. Compared with GAST and previous methods, the deep model in this paper showed higher classification accuracy. However, this article still needs a lot of modifications to be accepted.

My detailed comments are as follows:

1. For the section of Introduction:

·The unreliability of the GAST method is needed to explain.

·The two loss functions used in this paper are both improved on the basis of cross entropy function. "Use improved cross entropy loss function" or "Improve on cross entropy function" are suggested, instead of "propose a loss function".

2. For the section of Related Work:

·Some statements need to be modified to be more objective and rigorous. The last sentence of the first paragraph can be changed to "The domain adaptation on the point cloud has not been considered". For the statement of Qin et al.[13], it is suggested to specify the method to be used, the characteristics of the method, and finally the shortcomings, rather than saying "however“ at the beginning.

3. For the section of “Unsupervised Domain Adaptation for Point Cloud Classification”:

·For formula (1), comparison between the original formula in literature [14] and formula (1) is suggested. Also, the reasons for the inappropriateness needs to be explained.

·The model proposed in this paper is quite different from GAST. However, in Section 3.1, both the methods of GAST and proposed method are introduced. Although the principles in Section 3.1.1 are applicable to both, it is still recommended that GAST be introduced separately.

·For section 3.3, the effect after Cutting Plane Identification and the advantages of this method need to be described in detail.

·How to select and process the two parts of data, Pl and Pr, after cutting the point clouds? For the whole object, the two parts of the point cloud may be quite different (for most asymmetric objects/objects). Will it have a big impact on the experimental results?

·In addition, is the cut section information useful?

4. For the section of Experiments:

·For Section 4.1, the corresponding relationship between M10, S10 and S*10 datasets (so are the M11 & S*O11, S9 & S*O9) needs to be specified, as well as the rationality or reliability of these corresponding relationships. The above relationships are used in Section 4.2, but no proof of the relationship between the two datasets are introduced in the article. "

·For “ModelNet40 [28] is it a clerical error? If so, please modify it.

·The size of datasets used in this article is not described in this article.

·For Section 4.2, it is recommended to introduce the training process first and then the testing process.

·For  λvote mentioned above, why choosing the maximum value that λvote=1 in the first training? (Although several attempts have been made in Section 4.3.3)

·Training processes can be described in detail, including software and hardware conditions, as well as training and testing time.

·For Section 4.3.2, how to use the existing model (e.g. GAST) for testing? (e.g. all methods are trained and models are obtained under the experimental conditions in this paper) It should be stated that all methods are tested under the same experimental conditions to ensure the fairness of the results.

·How to improve the experimental process to reduce the confusion of similar objects in the model? If this aspect has been studied, it can be explained.

In a word, the methods in this paper provided higher accuracy of point cloud classification, and showed excellent performance in various self-supervised algorithms. However, the statement of the experimental process needs to be improved, and the method in this paper also has limitations. More experiments are needed to reduce the limitations.

These are all my comments on this article.

Reviewer 2 Report

The authors presented an interesting research work. However, the following additional considerations can be taken into account:

1. How the present research work differs in terms of accuracy from past works?

2. Graphical abstract can be added.

3. What is the limitations of the current work? How can it be overcome?

4. What is the future scope of this work? Can it be implemented in real-time?

Author Response

We underwent English revision using editing service.

Also, please see the attachment as reply.

Round 2

Round 3
